# The Mechanical and Thermoelectric Properties of Bi$_2$Te$_3$-Based Alloy Prepared by Constrained Hot Compression Technique

Dongsheng Qian [1,2], Ziyi Ye [1,2], Libo Pan [3], Zhijiang Zuo [3], Dongwang Yang [4,*] and Yonggao Yan [4]

1   School of Materials Science and Engineering, Wuhan University of Technology, Wuhan 430070, China; qiands@whut.edu.cn (D.Q.); yeziyi@whut.edu.cn (Z.Y.)
2   Hubei Engineering Research Center for Green Precision Material Forming, Wuhan 430070, China
3   School of Intelligent Manufacturing, Jianghan University, Wuhan 430056, China; panlb@jhan.edu.cn (L.P.); zuozj@jhan.edu.cn (Z.Z.)
4   State Key Laboratory of Advanced Technology for Materials Synthesis and Processing, Wuhan University of Technology, Wuhan 430070, China; yanyonggao@whut.edu.cn
*   Correspondence: ydongwang@whut.edu.cn

**Abstract:** This study proposes a constrained hot compression-molding (CHCM) technique for preparing Bi$_2$Te$_3$-based alloys. This method overcomes the problem of easy cleavage and destruction for the zone-melted Bi$_2$Te$_3$-based alloy, which is beneficial to improve the material utilization rate and thermoelectric devices yield in the commercial manufacturing process. The stress field distribution inside the CHCM specimen is explored via finite element analysis. The compressive strength of the CHCM sample is above 44 MPa, which is about 38% higher than that of the zone melting (ZM) material. Meanwhile, the CHCM sample shows a much lower electrical conductivity and thermal conductivity, but a higher Seebeck coefficient than that of the ZM sample, which is mainly due to the increase of the line defect concentration induced by the CHCM process. Finally, a maximum thermoelectric figure of merit (*ZT*) value of 0.6 was achieved for CHCM sample.

**Keywords:** Bi$_2$Te$_3$-based alloys; hot compression; thermoelectric properties; mechanical properties



## 1. Introduction

Thermoelectric (TE) material can realize the direct conversion from heat into electricity [1–4]. Because of the technique merits, such as local point cooling, free of noise and pollution [5], thermoelectric device finds broad applications in power generation and solid-state refrigeration [6–8]. The performance of TE materials is characterized by the figure of merit *ZT*, $ZT = \alpha^2 \sigma T / \kappa$, where $\alpha$, $\sigma$ and $\kappa$ are the Seebeck coefficient, electrical conductivity, absolute temperature and total thermal conductivity of the material, respectively. The electrical property parameter $\alpha^2 \sigma$ is also called the power factor [9–11].

Bi$_2$Te$_3$-based alloys exhibit excellent TE performance around room temperature [12–18]. Commercially available Bi$_2$Te$_3$-based materials are usually manufactured by the traditional unidirectional crystal growth methods, such as Bridgman or zone melting (ZM) techniques [19]. The maximum *ZT* value is about 1.0 at room temperature [20]. Bi$_2$Te$_3$ single crystal shows a hexahedral layered structure comprising of Te (1)-Bi-Te (2)-Bi-Te (1) layers. The atoms in each layer and the Bi/Te atoms within adjacent layers are ionically and covalently bonded, while the two penta-atomic layers are combined by the van der Waals force [21]. Therefore, the Bi$_2$Te$_3$ crystal is prone to cleavage along the (00*l*) plane perpendicular to the crystal c axis. The ZM ingots show high TE performance, while poor mechanical strength [22]. Although the Bi$_2$Te$_3$-based alloys with random crystal orientation show better mechanical performance yet sacrificing the TE properties [23].

Grain refining has been recognized as an effective method to improve the strength and toughness of materials [24–27], and various studies have shown that grains can be effectively refined under three-dimensional compressive stress [28,29]. This work aims

to obtain Bi$_2$Te$_3$-based alloys with refined grains and preferred orientation through a constrained hot compression-molding (CHCM) process, thus, to improve the TE and mechanical properties of Bi$_2$Te$_3$-based material simultaneously.

During the CHCM process, the crystal grains of Bi$_2$Te$_3$-based ZM ingot are damaged when retaining at a three-dimensional compressive pressure, thus inducing a high density of line defects inside the grains. These line defects no doubt violently scatter electrons and phonons and reduce the electrical conductivity and thermal conductivity. The line defects intertwine with each other, which hinder the slip of the dislocation to some extent and increase the strength of the bulk material.

## 2. Experiment Section

### 2.1. Material Synthesis

The schematic diagram of the CHCM method used in this work is shown in Figure 1. The high static pressure is loaded on the Bi$_2$Te$_3$-based alloys, which are wrapped in pure aluminum sleeve.

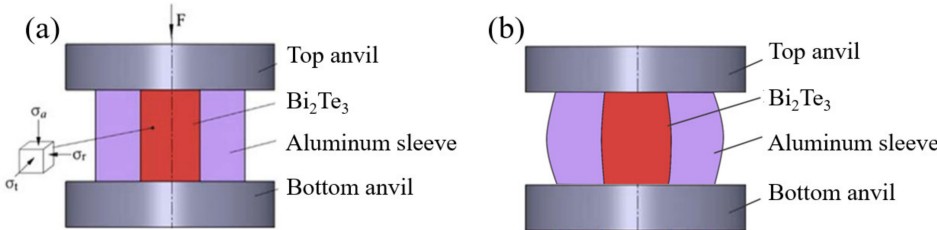

**Figure 1.** Schematic diagram of constrained hot compression molding (CHCM) technique: (**a**) before and (**b**) after compression.

The p-type Bi$_2$Te$_3$-based alloy (Bi$_{0.5}$Sb$_{1.5}$Te$_3$ ZM bar material) is wire-cut into a $\Phi15 \times 20$ mm$^3$ cylinder, of which the axis is the same with the ZM direction. The size of the outer aluminum sleeve is $(\Phi60 - \Phi15) \times 20$ mm$^3$ (Figure 2a). Some BN powders were spin-coated on the inner wall. The whole specimen was heat up to 714 K, and dwelt for 30 min, then loaded by a pressure of 1200 KN with a punching speed of 3 mm/s (Figure 2b) and dwelt for 4 h at 714 K.

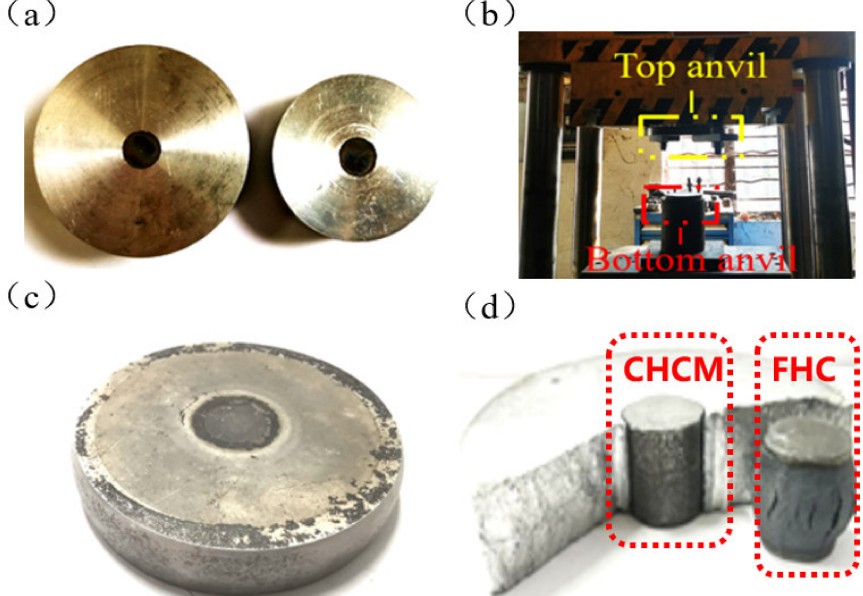

**Figure 2.** (**a**) Aluminum sleeve in CHCM experiment, (**b**) CHCM experiment (1200 KN press), (**c**) photo of mold after CHCM test, (**d**) comparison of the appearance between CHCM and FHC sample.

Figure 2c shows the photo of the mold after CHCM test, and Figure 2d displays the comparison of the appearance between the CHCM test sample and the free hot compression (FHC) test sample. It can be clearly seen that there are obvious drum-shaped and macroscopic cracks on the surface of the FHC sample, while the surface of the CHCM sample is much smoother with little surface cracks and better molding quality.

### 2.2. Structure Characterization and Property Measurement

The resulting cylindrical ingot was cut into $3 \times 3 \times 10$ mm$^3$ for electrical property (ZEM-3, UlvacRiko, Kanagawa, Japan) and $\Phi\,6 \times 2$ mm$^3$ for thermal diffusivity (LFA457, Netzsch, Bavaria, Germany) tests. The phase purity of all samples was inspected by X-ray powder diffraction (XRD) (Empyrean, CuK line, PANalytical, Almelo, Holland). Images of freshly fractured surfaces were taken by field emission scanning electron microscopy (FE-SEM) (SU8020, Hitachi, Tokyo, Japan). High-resolution transmission electron microscopy (HRTEM, JEM-2100F, JEOL, Tokyo, Japan) was used to determine the fine microstructures in the bulk materials.

## 3. Results and Discussion

### 3.1. Finite Element Simulation of CHCM Process

During the finite element analysis, the inner Bi$_2$Te$_3$-based alloy, the outer pure aluminum sleeve, the upper and lower molds are established, and the upper and lower molds are regarded as rigid bodies. Symmetrical simulation analysis method was used to simplify the operation process, and 1/4 of the 3D local aluminum sleeve model is deformed. It can describe the deformation of the whole 3D assembly in the deform post-processing module. The Bi$_2$Te$_3$-based alloy was set to plastic body, with the extrusion temperature of 714 K and the top anvil downward speed of 1 mm/s. The friction between the alloy and the aluminum was set as coulomb friction with a friction coefficient of 0.2.

Figure 3a shows the compression stress field distribution of the Bi$_2$Te$_3$ sample in a free state, i.e., the model does not contain an outer aluminum sleeve at this time. Figure 3b displays the point-tracking stress in the drum-shaped area. It can be seen that the outer of the Bi$_2$Te$_3$-based specimen is affected by the metal radial flow expansion, to withstand the circumferential direction tensile stress. In the free compression state, the material drum-shaped area is always in the tensile stress state, which would increase along with the pressure rising.

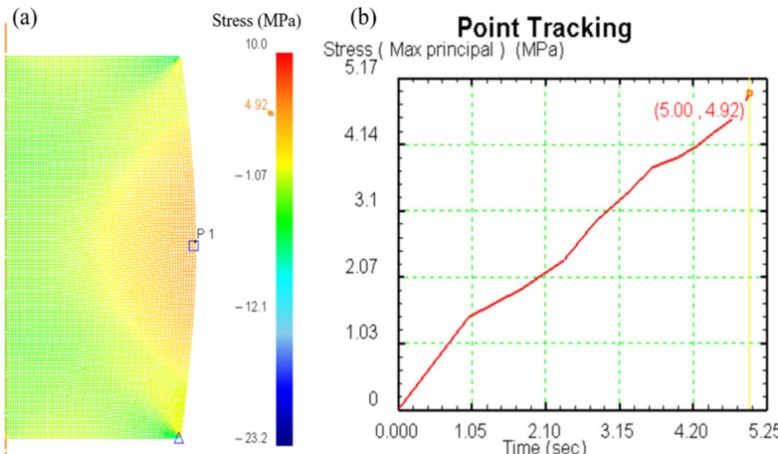

**Figure 3.** (**a**) Stress distribution of the Bi$_{0.5}$Sb$_{1.5}$Te$_3$ sample under free state hot compression condition, and (**b**) point tracking of the stress state at the apex of the drum-shaped area.

Micro-crack will germinate and propagate in the material with poor plasticity when the tensile stress exceeds the actual tensile strength, leading to poor molding quality and performance (Figure 2d). The outer aluminum sleeve ensures that the Bi$_2$Te$_3$-based alloy

specimen was uniformly deformed under three-dimensional compressive stress, so as to suppress the radial drum formation. Figure 4a shows the stress distribution of the aluminum sleeve in the CHCM process. It can be seen that the drum-shaped part is significantly reduced. The drum-shaped point tracking stress state shows that the drum-shaped area is always in the compressive stress state during the CHCM process (Figure 4b), which gradually increases as the time. Further, the closer to the core of the material, the greater the compressive stress. The CHCM process significantly changes the internal stress state of the inner specimen, with the stress state of the drum part of the material changing from tensive to compressive, so as to suppress the cracking of the material (Figure 2d).

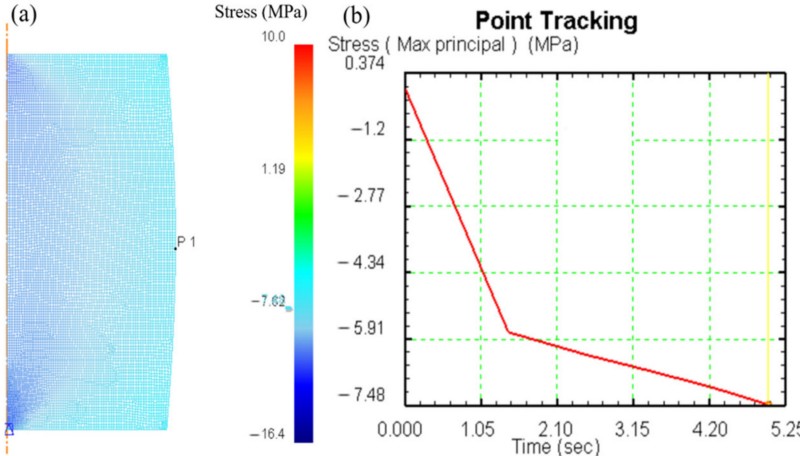

**Figure 4.** (**a**) Compression stress distribution of the aluminum sleeve in the CHCM process, and (**b**) point tracking of the stress state of at the apex of the drum-shaped area.

## 3.2. Structure and Performance of CHCM Material

Figure 5 shows the XRD patterns of CHCM samples parallel to the pressure direction under different deformation conditions at 714 K. It can be seen that all three samples are in single-phase structure of $Bi_{0.5}Sb_{1.5}Te_3$. For the 30% deformation sample, the diffraction peak intensity of the crystal plane closely related to the (00$l$) direction is significantly higher than that on other crystal planes. It means that the crystal grains are preferred arranged in a certain direction under the three-dimensional force.

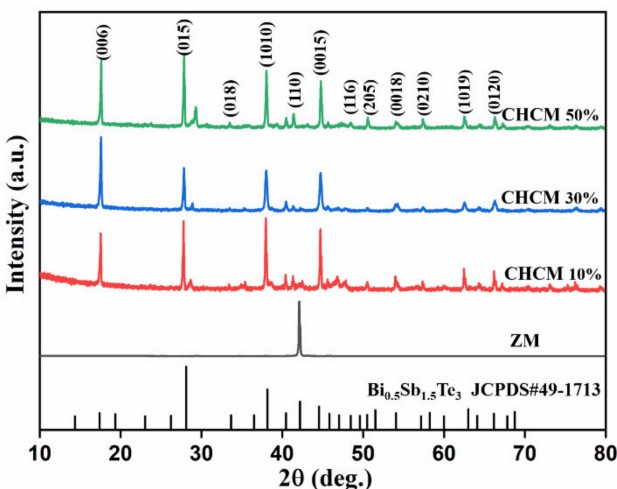

**Figure 5.** XRD patterns of $Bi_{0.5}Sb_{1.5}Te_3$ samples by ZM and CHCM under different deformations.

Scanning electron microscope (SEM) images of the samples prepared under different deformations (10 and 30%) are shown in Figure 6. It can be seen from Figure 6a that the 10% deformation CHCM sample exhibits lamellar structures with obvious texturization.

The growth along a certain direction is obvious. The failure of the sample starts from the cleavage plane between the lamellae, then propagates layer by layer, showing a typical trans-crystalline fracture behavior (Figure 6b). There are a small number of voids among large crystal grains, so the bulk material is not completely dense.

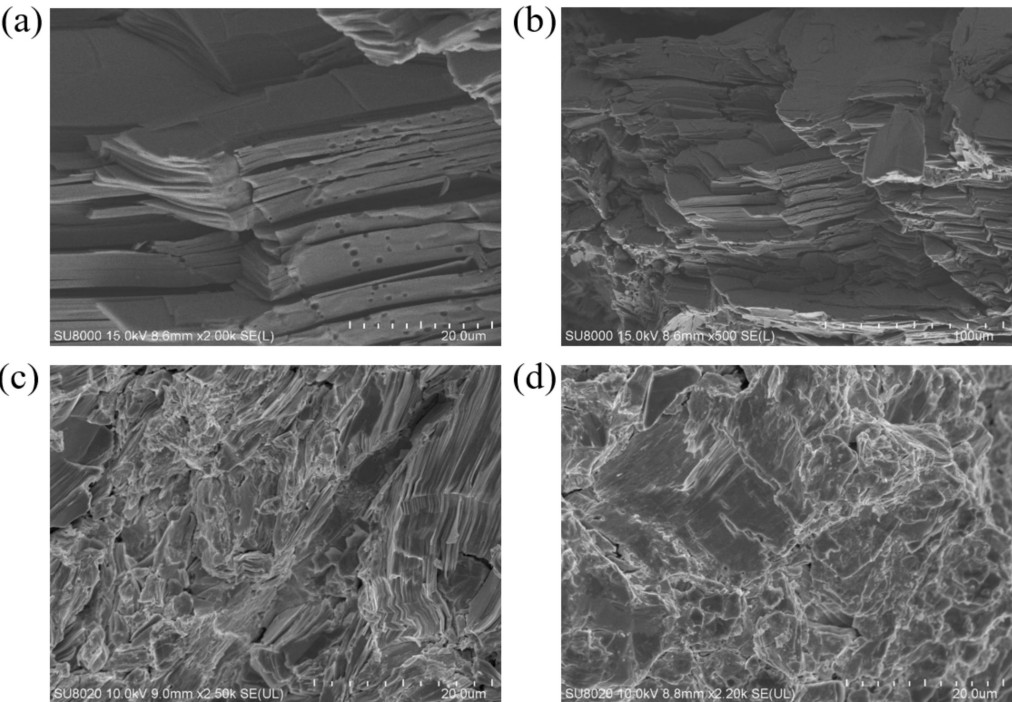

**Figure 6.** (**a**,**b**) SEM image of the 10% deformation CHCM specimen; (**c**,**d**) SEM images of the 30% deformation CHCM specimen, (**c**) high magnification image with the direction parallel to the load axis, (**d**) high magnification image with the direction perpendicular to the load axis.

Because the $Bi_2Te_3$–based alloy has a hexahedral layered structure, the Bi atoms and Te atoms in the c-axis direction are arranged cyclically in the manner of Te(1)-Bi-Te(2)-Bi-Te(1), and the atoms within these penta-atomic-layers are ionically and covalently arranged, and the combination of Te(1)-Te(1) between these penta-atomic-layers through van der Waals force leads to its anisotropy [22–25]. It can be observed from Figure 6c,d that there is a certain degree of anisotropy in the direction parallel and perpendicular to the pressure, which is consistent with the above XRD results. It proves that the local preferred orientation structure could be formed during the CHCM process.

In order to further analyze the preferred orientation of the grain growth of the sample, the orientation factor of the sample can be calculated by the Lotgering method [30]. The definition of orientation factor is:

$$F = \frac{P - P_0}{1 - P_0} \tag{1}$$

$$P = \frac{\Sigma I(00l)}{\Sigma I(hkl)} \tag{2}$$

$$P_0 = \frac{\Sigma I_0(00l)}{\Sigma I_0(hkl)} \tag{3}$$

$P$ represents the ratio of the sum of $(00l)$ diffraction intensities to the sum of all the characteristic peak intensities in the XRD pattern of CHCM sample. $P_0$ here represents the same ratio of the ZM material. Table 1 lists the orientation factors of the CHCM samples calculated according to the above formula. From the data in Table 1, it can be seen that the CHCM samples have more significant grains texturization than HP samples. The directional growth trend is much more obvious for 30% deformation CHCM sample.

**Table 1.** Orientation factors of CHCM samples with different deformations and HP samples.

| Samples | 10% CHCM | 30% CHCM | 50% CHCM | HP [24] |
|---|---|---|---|---|
| Orientation Factor | 0.2716 | 0.4049 | 0.2473 | 0.12 |

Figure 7a depicts the fine microstructure of the CHCM sample with 30% deformation, showing a large number of line defects inside the grains. Figure 7b is the enlarged view image of the yellow rectangle area. The obvious contrast means large strain around the line defect, which would significantly scatter the electrons and acoustic phonons.

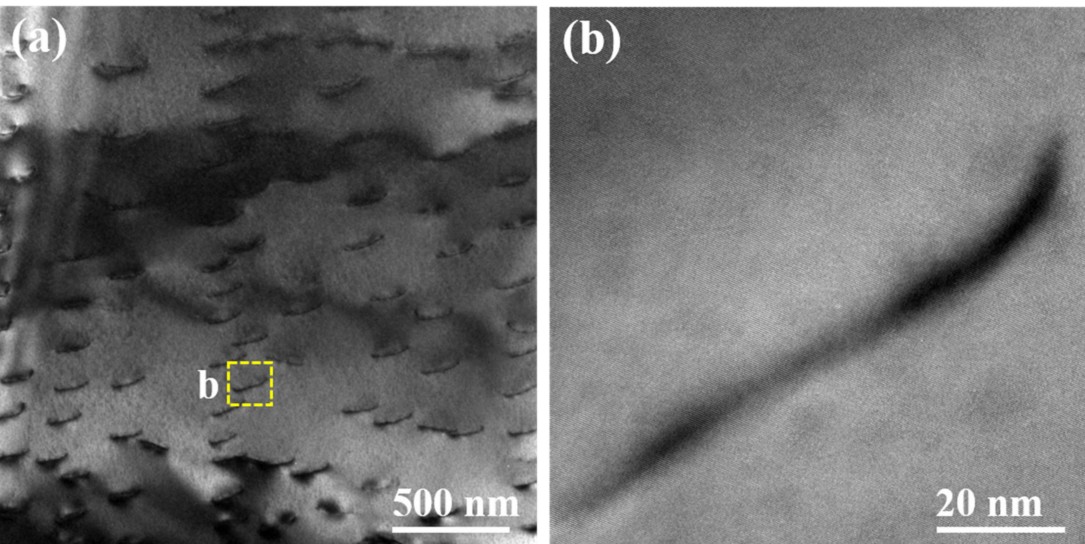

**Figure 7.** Fine microstructure of the CHCM sample with 30% deformation. (**a**) Low-magnification TEM images showing a high density of line defects, (**b**) the enlarged view image of the yellow rectangle area in (**a**).

Thermoelectric properties perpendicular to the pressure direction were measured for the CHCM sample with 30% deformation, and the ZM sample acts as a comparison, as shown in Figure 8. As the temperature increasing, the electrical conductivity of all samples decreases, exhibiting a metallic conduction behavior (Figure 8a). The electrical conductivity of the CHCM sample is significantly lower than that of ZM sample due to a reduced carrier mobility. Figure 8b displays the temperature dependent Seebeck coefficient of CHCM sample with 30% deformation and ZM sample. The CHCM sample has the largest Seebeck coefficient of 240 μV/K near room temperature, and the Seebeck coefficient first decreases slowly as the temperature rising, then decreases rapidly after 400 K due to intrinsic excitation. Compared with ZM sample, the Seebeck coefficient of CHCM sample is significantly improved below 450 K, which may be related to the fact that the anti-site defects generated after deformation in the CHCM sample reduce the carrier concentration of the bulk material [31–34]. As a result, the power factor of CHCM sample reaches $2 \times 10^{-3}$ W/mK$^2$, which is much lower than that of ZM sample (Figure 8c). It means that the carrier concentration has deviated from the optimum value.

Figure 8d displays the comparison of total thermal conductivity between ZM and CHCM samples. The total thermal conductivity of CHCM sample is much lower than that of ZM sample due to its reduced electrical conductivity. The temperature-dependent lattice thermal conductivity calculated by directly subtracting electron thermal conductivity and bipolar thermal conductivity from the total thermal conductivity is displayed in Figure 8e. The CHCM process significantly reduces the lattice thermal conductivity through line defects generating and strain-phonon scattering enhancing. Finally, the figure of merit *ZT* values could be calculated and shown in Figure 8f. A maximum *ZT* value of 0.6 is achieved around room temperature. Although the TE properties are deteriorated in the end, proper

adjustment of the composition may help further optimize the TE performance based on the beneficial effect of CHCM technique in reducing the thermal conductivity. The *ZT* value is higher as those of the samples fabricated by BMA-HP [23] and MA-PAS methods [35], CHCM samples exhibited preferred orientation, which are beneficial to thermoelectric and mechanical performance.

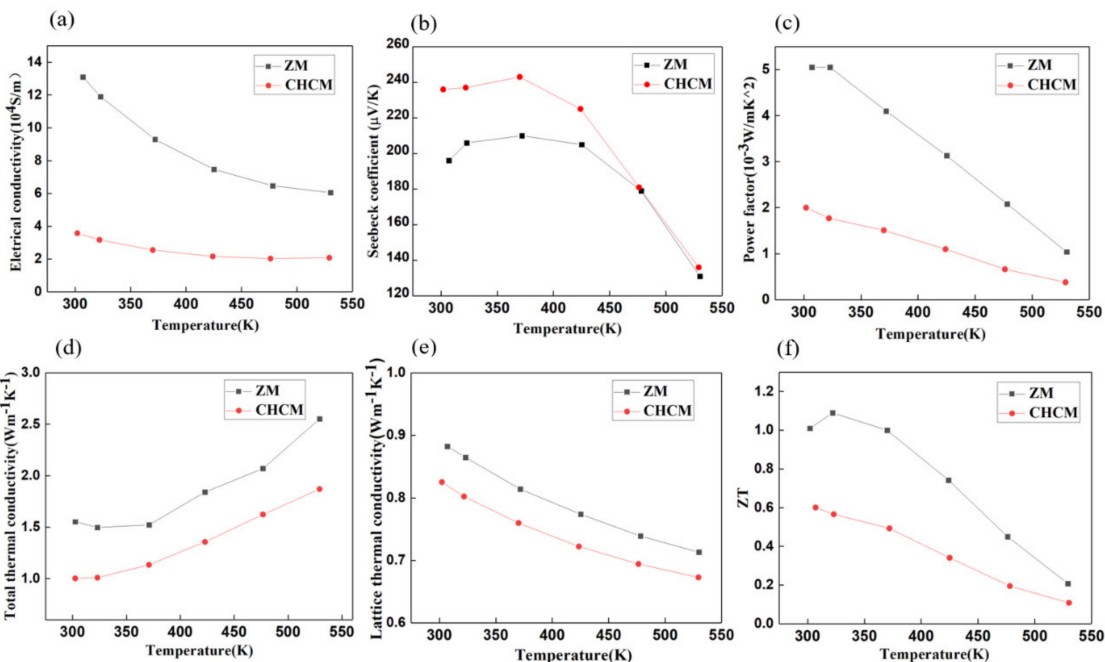

**Figure 8.** Thermoelectric properties of the CHCM sample with 30% deformation and the ZM sample. (**a**) Electrical conductivity, (**b**) Seebeck coefficient, (**c**) power factor, (**d**) total thermal conductivity, (**e**) lattice thermal conductivity, (**f**) *ZT* value.

During the CHCM process, the material undergoes partial melting, directional arrangement under three directional compressive stress, crystal growth and recrystallization. This process would promote the line defects intertwining with each other and also increase the relative density of the CHCM sample, thus improve the mechanical properties simultaneously.

Table 2 shows the compressive strength of the CHCM samples with different deformation constraints. Compared with the ZM samples, the compressive strength of the CHCM samples has been greatly improved, and the mechanical properties of the samples are improved with the deformation degree increasing. Specifically, the compressive strength of the CHCM sample with 50% deformation reaches 67.2 MPa, an increasement of 175% comparable to ZM sample.

**Table 2.** Comparison of compressive strength of ZM and CHCM samples with different deformations.

| Samples | ZM | 10% CHCM | 30% CHCM | 50% CHCM |
|---|---|---|---|---|
| Compressive strength (MPa) | 24.4 | 43.2 | 58.5 | 67.2 |

## 4. Conclusions

In the present study, a series of $Bi_2Te_3$-based alloys were prepared via using the CHCM technique. A CHCM model is constructed based on the three-directional compressive stress state. The compressive strength of the CHCM sample is above 44 MPa, which is about 38% higher than that of the zone melting (ZM) material. Meanwhile, the CHCM sample shows a much lower electrical conductivity and thermal conductivity, but a higher Seebeck coefficient than that of the ZM sample, which is mainly due to the increase of the line defect

concentration induced by the CHCM process. Finally, a maximum *ZT* value of 0.6 was achieved for CHCM sample.

**Author Contributions:** Conceptualization, D.Q. and D.Y.; Methodology, D.Q., Y.Y. and D.Y.; Software, Z.Y., Z.Z. and D.Y.; Validation, D.Q., Z.Y. and D.Y.; Formal analysis, Z.Y.; Investigation, Z.Y.; Resources, D.Q., Y.Y. and D.Y.; Data curation, D.Q., Y.Y. and D.Y.; Writing—original draft preparation, Z.Y.; Writing—review and editing, D.Q., Y.Y. and D.Y.; Visualization, D.Q., L.P. and Z.Z.; Supervision, D.Q., Z.Z. and L.P.; Project administration, D.Q. and L.P.; Funding acquisition, D.Q. and Y.Y. All authors have read and agreed to the published version of the manuscript.

**Funding:** This research was funded by Innovative Research Team Development Program of Ministry of Education of China (No. IRT_17R83), 111 Project (B17034), and Wuhan Frontier Project on Applied Research Foundation (Grant No. 2019010701011405).

**Data Availability Statement:** Not applicable.

**Acknowledgments:** The authors are grateful for the access to testing facilities at State Key Laboratory of Advanced Technology for Materials Synthesis and Processing at Wuhan University of Technology, China.

**Conflicts of Interest:** The authors declare that they have no known competing financial interests or personal relationships that could have appeared to influence the work reported in this paper.

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
