# Peer review of "The Mechanical and Thermoelectric Properties of Bi2Te3-Based Alloy Prepared by Constrained Hot Compression Technique"

_metals, doi:10.3390/met11071060_

Round 1

Reviewer 1 Report

The authors developed the hot compression molding technique for prepare Bi-Te-based thermoelectric materials in an effort to improve the mechanical reliability. Although the obtained zT is lower than that of a commercial ingot, the proposed processing technology is worthy in fabrication of thermoelectric materials with controlled transport properties. Several minor issues should be properly addressed before publication.

1) XRD pattern for ZM sample is required in Fig. 7.

2) zT of Bi-Te-based alloys strongly depends on their composition. Detailed information about compositions of ZM and CHCM samples should be provided. And the related discussion for the difference in zT between ZM and CHCM samples is required.

3) In Fig. 10(c), calculated lattice thermal conductivity might include the bipolar contribution. Estimation of the lattice thermal conductivity after substract of bipolar thermal conductivity is required.     

Reviewer 2 Report

The authors report on their studies on Bi2Te3 based alloys. Their main focus is on the samples prepared by constrained hot compression molding (CHCM) and the improvement of the sample quality for materials utilization and thermoelectric devices. The compressive strength of these samples is compared to samples prepared by zone melting (ZM). Although the maximum ZT (figure of merit for thermoelectric) is lower for the CHCM samples than for the ZM samples the stress properties are for applicational usage enhanced.

Reviewer 3 Report

This manuscript, entitled „The Mechanical and Thermoelectric Properties of Bi2Te3-based Alloy Prepared by Constrained Hot Compression Technique’’ is an interesting, with  good scientific level that fits well within the scope of this Journal.

A good synthesis of the literature offering an overview of the evolution researches in the area.

However, some points need to be addressed prior to publication of this manuscript. My comments/suggestions are given:

  1. For XRD analysis, the card used to identify the peaks or at least a bibliographic reference must be specified.
  2. The legend of figure 8 must be reformulated because it is not very well understood. I believe that c and d are SEM images of the 30% deformation CHCM specimen, in which (c) parallel to the load axis and (d) perpendicular to the load axis.
  3. Figure 9 is an SEM micrograph? Specify! Figure 9b should be replaced with a clearer one, if possible.
  4. Figure 10 should be clearer because the values are harder to see.

Author Response

This manuscript is a resubmission of an earlier submission. The following is a list of the peer review reports and author responses from that submission.

Round 1

Reviewer 1 Report

Authors have added some data on zT and thermal conductivity which improves the paper a bit. However, as it was pointed out in the first review, the merit of the paper is not justified. The lattice part of the thermal conductivity is same as that of the zone melted material which does not justify the one of the main idea of paper "grains refining 
could also increase the interface concentration, enhance the 
phonon scattering, and thus reduce the lattice thermal 
conductivity".  Thus, additional explanation or experiment are needed to justify the design of the paper. The zT of the material after processing is reduced to almost half of the original value and the mechanical strength is somewhat improved. With this result, it is hard to claim that the material has been improved in thermoelectric application. I recommend authors to improve/optimize the process and achieve at least a comparable zT to zone melted material before submitting the paper again. It can not be accepted for publication in the current form.  

Reviewer 2 Report

The authors resolved the issues that the referee raised before and revised the manuscript accordingly. So I recommend this manuscript is suitable to be published as in this form.

Reviewer 3 Report

Review on “The Mechanical and Thermoelectric Properties of Bi2Te3-based Alloy Prepared by Constrained Hot Compression Technique” submitted to Metals.

This paper reports investigation into processing of Bi2Te3-based materials with constrained hot compression-molding. Compared to zone melted materials, the present materials exhibit significantly higher compressive strength of 44 MPa. Much lower electrical conductivity and lattice thermal conductivity are obtained with higher Seebeck coefficients which is explained by reduction of the carrier concentration and increase of the defect concentration. I think this is an interesting work with useful finite element simulation of the CHCM process, which tries to address the mechanical characteristics issues of Bi2Te3. I can recommend publication with the following minor points addressed. First importantly, Bi2Te3 is by far the most studied thermoelectric material system, and in addition to the specific papers given, I think some overview and notable advancement references on Bi2Te3 should also be given to be helpful to readers in addition to [10], like J. Pei, et al. National Science Review, Volume 7, Issue 12, December 2020, Pages 1856–1858, https://doi.org/10.1093/nsr/nwaa259, T. Fang et al., Adv. Funct. Mater. 29 (2019) 1900677, W. Liu et al., Energy Environ. Sci., 2013, 6, 552, A. Pakdel et al., J. Mater. Chem. A, 2018, 6, 21341-21349. Yufei Liu, et al., Scripta Materialia, 111, 2016, 39-43, doi: 10.1016/j.scriptamat.2015.06.031, S. Grasso, et al., Journal of Material Chemistry C (2013), 1, 2362-2367, W. Liu, et al., Energy Environ. Sci., 2013,6, 552-560, etc. And regarding the importance and applications of thermoelectrics, [1] to [6], for Bi2Te3 especially, energy harvesting IoT application is an especially important application for TE at room temperature and should be added; L. E. Bell, Science 321, 1457 (2008), I. Petsagkourakis et al., Sci. Tech. Adv. Mater., 19, 836-862 (2018), etc.
